# Quantum chaos and the arrow of time

**Nilakash Sorokhaibam**[1*]

**1** Department of Physics, Tezpur University, Tezpur, 784028, Assam, India.

⋆ phy_sns@tezu.ernet.in

## Abstract

Classical physics possesses an arrow of time in the form of the second law of thermodynamics. But a clear picture of the quantum origin of the arrow of time has been lacking so far. In this letter, we show that an arrow of time arises naturally in quantum chaotic systems. We show that, for an isolated quantum system which is also chaotic, the change in entropy is non-negative when the system is perturbed. At leading order in perturbation theory, this result follows from Berry's conjecture and eigenstate thermalization hypothesis (ETH). We show that this gives rise to a new profound constraint on the off-diagonal terms in the ETH statement. In case of an integrable system, the second law does not hold true because the system does not thermalize to a generalized Gibbs ensemble after a finite perturbation.

## 1 Introduction

All physical systems around us become more and more disordered as time progresses unless an outside agent spends energy and record information for the upkeep of the system. This phenomenon has a beautiful statistical reasoning. There simply are too many possible disordered states compare to the number of possible ordered states. So, the system ends up most likely in a disordered state [1]. Entropy is the quantity which measures the extent of disorder.

In terms of entropy, this general observation gives rise to the second law of thermodynamics. It states that the entropy of an isolated system does not decrease. This is Planck's statement of the second law of thermodynamics [2]. It gives rise to an arrow of time (called *Time's arrow* in [3]). These results are familiar in the realm of classical physics. But our physical world is governed by quantum mechanics at microscopic level and so far a complete understanding of the quantum version of the second law is lacking.

In this work, we show that an arrow of time arises naturally in an isolated quantum system which is chaotic. Quantum chaos in many-body systems and the related question of thermalization of these systems have been areas of intense research in the last few decades [4]. Traditionally, quantum chaos is identified from the study of energy level statistics [5], while thermalization in a quantum system follows from eigenstate thermalization hypothesis (ETH) [6,7]. The main idea of ETH comes from Berry's conjecture [8] which states that energy eigenstates of quantum chaotic systems are thermal in an observational sense. ETH explains how observables thermalize in a quantum system. So, it is the route to statistical mechanics for quantum systems. ETH has been shown to hold true for physically sensible observables of quantum chaotic systems while it does not hold true for quantum integrable systems. So for the rest of the paper, we will assume that quantum chaos and ETH imply each other.

At leading order in time-dependent perturbation theory, the second law follows from Berry's conjecture and ETH. We find that a naive substitution of ETH is not enough. A closer examination led us to a new profound constraint on the ETH statement. We also study chaotic systems numerically. We find that the higher order terms in the perturbation theory are suppressed for reasonable values of the perturbation strength. We also study large perturbations numerically. We found that the change in energy is dictated by the second law. Even though the energy jumps the infinite temperature region, the change in entropy is still positive.

It has previously been shown that the change in the *diagonal* entropy is non-negative under doubly stochastic evolution starting from a *passive* density matrix [9]. This follows from the convex nature of the function $x \log x$. This result is true even for integrable systems. But integrable systems do not thermalize after a finite perturbation. In the present work, we will be working with pure states which is the most appropriate approach for an isolated system. In the spirit of Berry's conjecture and ETH, we will be working with individual energy eigenstates.

Just as we cannot keep track of the positions and momenta of all gas molecules in the classical theory of gas, we cannot keep track of the complex coefficients of the exponentially large number of energy eigenstates of a quantum system. So in quantum mechanics, we keep track of only certain observables which can be easily measured or calculated. These can be local operators or global operators but they are few-body operators. If we observe that these observables thermalize then we say that the system thermalizes. ETH is a criteria for thermalization of such observables so that their long time expectation values can be described by quantum statistical mechanics. Inevitably, the quantum operators corresponding to these observables are hermitian and non-fermionic in nature.

Consider an operator $\mathcal{O}$ which corresponds to one such observable. ETH states that the matrix elements $\mathcal{O}_{mn}$ in energy eigenbasis are of the form

$$\langle m|\mathcal{O}|n\rangle \equiv \mathcal{O}_{mn} = \mathcal{O}(\bar{E})\delta_{mn} + e^{-S(\bar{E})/2}f(\bar{E}, \omega)R_{mn} \tag{1}$$

where $|m\rangle$ and $|n\rangle$ are energy eigenstates with energies $E_m$ and $E_n$ respectively and $\bar{E} = (E_m + E_n)/2$, $\omega = E_m - E_n$. $S(\bar{E})$ is the entropy at energy $\bar{E}$. $\mathcal{O}(\bar{E})$ is equal to the expectation value of $\mathcal{O}$ in the microcanonical ensemble at energy $\bar{E}$ or other ensembles by the equivalence of ensembles. $\mathcal{O}(\bar{E})$ and $f(\bar{E}, \omega)$ are smooth functions of their arguments. $f(\bar{E}, \omega)$ is taken to be a real and positive function. $f(\bar{E}, \omega)$ is of the order of 1 for a band $|\omega| < W/2$ [10] and falls exponentially for large $|\omega|$ [11]. $R_{mn}$ are pseudo-random variables with zero mean and unit

variance [12]. $\mathcal{O}$ is hermitian and hence

$$\mathcal{O}_{nm} = \mathcal{O}_{mn}^*, \; R_{nm} = R_{mn}^*, \; f(\bar{E}, -\omega) = f(\bar{E}, \omega) \tag{2}$$

In this work, we found that the second law of thermodynamics enforces that

$f(\bar{E}, \omega)$ *is a monotonically increasing function of* $S(\bar{E})$ *for* $|\omega| \gtrsim W/2$.

We will find that, in general, $f(\bar{E}, \omega)$ is a monotonically increasing function of $S(\bar{E})$ even inside the band $|\omega| < W/2$. The $\bar{E}$ dependence of $f(\bar{E}, \omega)$ has never been studied in detail in previous studies of ETH and quantum chaos. $\bar{E}$ is usually used only to fix $S(\bar{E})$ and the effective temperature. In this work, we are looking for unifying features of quantum chaos and ETH with respect to the second law of thermodynamics. So, we will not be concerned with minute differences between different ensembles when working with finite systems or other controversial cases.

In section 2, we will discuss different manifestations of the arrow of time. We will also write down the statement of the second law that we will be concentrating on. Section 3 describes the physical set-up and the analytic results obtained using time-dependent perturbation theory. Numerical results are presented in section 4. Section 5 covers higher order terms in the perturbation theory. Section 6 is conclusions. Numerical results of large perturbations are also presented in section 6. By perturbation, we mean disturbing the system, whereas perturbation theory means time-dependent perturbation theory of quantum mechanics.

## 2 Two manifestations of the arrow of time

The arrow of time has two different manifestations. The first manifestation is thermalization of non-equilibrium excited states which are typical [10]. If the statement of ETH is satisfied then the system thermalizes to the thermal state specified by the fixed conserved charges in the long time limit. This scenario is most popularly studied as quantum quenches where the non-equilibrium state is prepared by changing the Hamiltonian of the system [13, 14]. The initial Hamiltonian and the final Hamiltonian are different.

One subtlety when preparing states using quantum quenches is that even integrable systems thermalize to generalized Gibbs ensembles (GGEs) in the large system size limit, if the initial state has no long range correlation. Thermalization of quantum integrable systems after quantum quenches has been a topic of great interest in the last two decades [13–17]. The non-equilibrium states are generalized typical states taking into account all the other conserved charges of the integrable system. This result simply follows from central limit theorem and the local nature of the Hamiltonians [18]. The difference between chaotic systems and integrable systems arises when we consider finite perturbations for a finite duration of time (also called *bump quenches* in [19] and *critical to critical quench* in [17]). The initial Hamiltonian and the final Hamiltonian are same. A chaotic system still thermalizes after such finite perturbations [19] while integrable systems do not thermalize [17]. The effective temperature and other chemical potentials become imaginary. So in this manner, finite perturbations or bump quenches bring out the stark contrast between non-equilibrium dynamics of quantum chaotic systems and quantum integrable systems.

The second manifestation of the arrow of time is the quantum version of the second law of thermodynamics. Consider a quantum chaotic system which is initially in a thermal state (a pure state but thermalized). If we perturb the system, the system would be taken out of equilibrium but eventually the system would return back to thermal equilibrium. The second law would imply that the change in entropy ($\Delta S$) is non-negative. We will show that this is

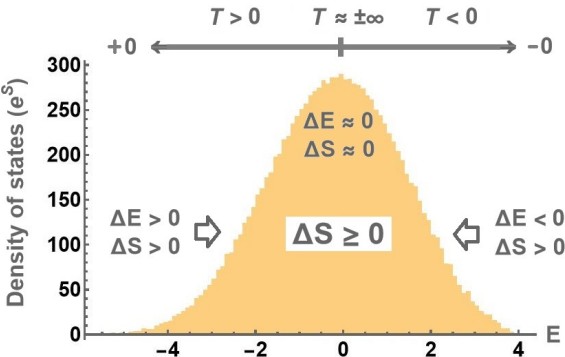

Figure 1: The density of states ($e^S$) versus energy ($E$) plot for the chaotic XXZ spin chain. $T$ is effective temperature if we are working with energy eigenstates. Perturbation of the system from a thermalized state always leads to a non-negative change in the entropy $\Delta S \geq 0$.

indeed the case for a quantum chaotic system. We show this by keeping track of the change in total energy of the system ($\Delta E$). The change in energy is positive if the initial temperature of the system is positive. If the system was initially at infinite temperature then the change in the energy is zero.

$$\Delta E = T \Delta S \tag{3}$$

where $\Delta E$ and $\Delta S$ are changes in energy and entropy. We have set the Planck constant $\hbar = 1$ and the Boltzmann constant $k_B = 1$. As mentioned above, we will be working with pure states. The effective temperature and entropy of the pure state are calculated using equivalence of ensembles. Basically we calculate these quantities from the canonical ensemble having the same energy as the pure state. In the rest of the paper, $T$ denotes the effective temperature of a pure state. For systems with finite dimensional Fock space, if the initial temperature is negative, the change in energy is negative and the change in entropy is positive again. Figure 1 is the schematic diagram of this result. One can also consider changes in other conserved charges by considering the change in the expectation value of an effective Hamiltonian which include the conserved charges with appropriate values of the chemical potentials. Our result in terms of the change in energy is the Kelvin's form of the second law of thermodynamics [20].

## 3   Set-up and analytic results

Consider a system with the time-independent Hamiltonian $H_0$ and initially in a state $|\psi\rangle$. The system is perturbed by a time-dependent term $\lambda(t)\mathcal{O}$ where $\mathcal{O}$ is a non-fermionic, hermitian operator. The total Hamiltonian is

$$H(t) = H_0 + \lambda(t)\mathcal{O} \tag{4}$$

The perturbation is for a finite duration of time, say, from time $-d$ to $d$. The source $\lambda(t)$ is real and can be non-zero only for $-d < t < d$. This perturbing term is not necessarily small but it also cannot be arbitrarily big. The change in energy of the system is

$$\Delta E = \langle \psi(t_f)|H_0|\psi(t_f)\rangle - \langle \psi(t_i)|H_0|\psi(t_i)\rangle \tag{5}$$

where $t_i < -d, d < t_f$. Our main goal is to show that $\Delta E$ is positive (negative, zero) if the initial effective temperature is positive (negative, infinite) when $H_0$ is chaotic and $\mathcal{O}$ satisfies ETH. This would imply that $\Delta S \geq 0$. The initial state is taken to be an energy eigenstate $|n\rangle$.

Using the Dyson series expansion of the time evolution operators, we can perform an expansion of $\Delta E$ in powers of $\lambda(t)$. We will concentrate on the leading term which is given by

$$\Delta E = \sum_m \Delta E_m = \sum_m (E_m - E_n)|\tilde{\lambda}(E_m - E_n)|^2 |\mathcal{O}_{mn}|^2 + O(\lambda^3) \tag{6}$$

where $\tilde{\lambda}(\omega) = \tilde{\lambda}(-\omega)^*$ is the Fourier transform of $\lambda(t)$. A naive substitution of the ETH expression results in

$$\Delta E = \int_{-\infty}^{\infty} d\omega\, \omega f(\bar{E}, \omega)^2 = 0 \tag{7}$$

because $f(\bar{E}, \omega)$ is a symmetric function of $\omega$. This naive substitution ignores the $\bar{E}$ variation of $|\mathcal{O}_{mn}|^2$. This naive substitution is routinely used to calculate correlation functions and study chaos and non-equilibrium dynamics (for example see [10]). But this cannot be true unless $E_n$ corresponds to infinite effective temperature. Now, this term actually is the linear response term and can be rewritten as

$$\Delta E = \frac{1}{2\pi} \int_0^{\infty} d\omega\, \omega\, |\lambda(\omega)|^2 A(\omega) \tag{8}$$

where $A(\omega) = -2\operatorname{Im} G_R(\omega)$ is the spectral function. $G_R(t, t') = -i\theta(t - t')\langle [\mathcal{O}_I(t), \mathcal{O}_I(t')]\rangle$ is the retarded Green's function where $\mathcal{O}_I(t)$ is the perturbing operator in the interaction picture. For a non-fermionic operator, the above expression is positive (negative or zero) in a thermal state of positive (negative or infinite) temperature. Invoking Berry's conjecture and ETH, we infer that the leading term in (6) and the thermal value should match. The difference will be fluctuations suppressed by exponentials of the entropy. Otherwise, one would be able to differentiate between an energy eigenstate and the corresponding thermal state by performing a small perturbation experiment and measuring the change in energy. On the other hand, this proposal would not hold true for integrable systems.

The naive substitution of ETH in (6) goes wrong in the details involving the integration (or sum) contour in the $\bar{E} - \omega$ plane. One has to be careful about the $\bar{E}$ variation. Figure 2 is an illustration of the $\bar{E} - \omega$ plane. The direction of $\omega$ and $\bar{E}$ axes are along the diagonals. The red line is the integration contour where we have consider only the values close to the $\bar{E}$ axis since $f(\bar{E}, \omega)$ is exponentially suppressed for large $|\omega|$. Using the symmetric nature of $f(\bar{E}, \omega)$ as a function of $\omega$, the right half of the contour can be rotated by $90°$ clockwise. The integration along the two segments will cancel if not for the $\bar{E}$ variation which is represented by the blue line. So, the $\bar{E}$ dependence of the integrand makes the total integral match with the thermal expectation value. This suggests that $f(\bar{E}, \omega)$ is a monotonically increasing function of $S(\bar{E})$ for $|\omega|$ greater than a certain value. Small $\omega$ region does not contribute significantly to the integral due to the $\omega\ (= E_m - E_n)$ factor in (6).

$f(\bar{E}, \omega)$ is a constant function in random matrix theory (RMT). It is well known that the long time physics from the band $|\omega| < W/2$ is governed by RMT. So, we expect that our proposal holds true for $|\omega| \gtrsim W/2$. In the next section, we will provide numerical evidences supporting this argument. We will find that the contribution to $\Delta E$ from small $|\omega| < W/2$ fluctuates. We will also find that $f(\bar{E}, \omega)$ is not a monotonically increasing function of $S(\bar{E})$ for particular values of $|\omega| < W/2$. Nevertheless, we will find that, in general, $f(\bar{E}, \omega)$ is a monotonically increasing function of $S(\bar{E})$ even for $|\omega| < W/2$.

To summarize, our two analytic results are

1. The change in energy at leading order starting from a single energy eigenstate should match with the value calculated from the corresponding thermal state.

2. $f(\bar{E}, \omega)$ is a monotonically increasing function of $S(\bar{E})$ for $|\omega| \gtrsim W/2$.

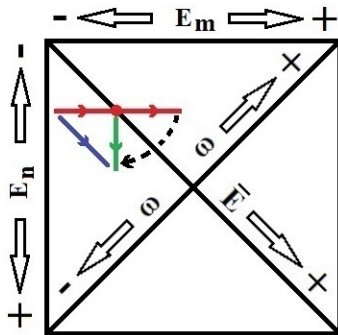

Figure 2: $\bar{E} - \omega$ plane. The red line is the integration contour of (6). The right half of the contour can be rotated by **90°** clockwise. The integral would be zero if not for the $\bar{E}$ variation which is represented by the blue line.

Again, these two inferences arise from Berry's conjecture and ETH statement. In the next section, we will consider some specific chaotic models and provide supporting numerical results.

## 4   Numerical results

We consider the XXZ spin chains with open boundary condition. With a large next-to-nearest neighbour interaction, the system is chaotic [21]. We consider system of size $L$ with the number of up-spins $N$. The system is perturbed using the global spin-current operator which preserves the total magnetization.

$$H_0 = \sum_{i=1}^{L-1} \left[ J_{xy} \left( S_i^x S_{i+1}^x + S_i^y S_{i+1}^y \right) + J_z S_i^z S_{i+1}^z \right]$$
$$+ \sum_{i=1}^{L-2} J_z' S_i^z S_{i+2}^z \tag{9}$$

$$\mathcal{O} = \sum_{i=1}^{L-1} J_{xy} \left( S_i^x S_{i+1}^x - S_i^y S_{i+1}^y \right) \tag{10}$$

We take $J_{xy} = 1$ and $J_z = J_z' = 0.5$ [1]. Figure 3 are the plots of the leading term in $\Delta E$ after a unit delta function perturbation $\lambda(t) = \delta(t)$ in each of the energy eigenstates. In the chaotic system, the leading term starting from different energy eigenstates match the thermal values considering both canonical and microcanonical ensembles. As one expects, the matching is better for larger systems. It is remarkable that the energy eigenstates of the chaotic $L = 12, N = 5$ system still shows thermal behaviour for the leading term of $\Delta E$ even though the dimension of the Fock space is only 792. For thermalization studies of chaotic systems, one usually need Fock space with dimensions of the order of **30000** [22, 23]. In case of the integrable system, the leading term starting from different energy eigenstates wildly vary compare to the thermal values. We will work with $L = 16, N = 7$ for the rest of the numerical results.

The left panel in Figure 4 is the plot of $\Delta E_m$ (as defined in (6)) as a function of $E_m$ for a fixed $E_n$ with effective temperature $T = 2$. We can see the exponential fall-off of $f(\bar{E}, \omega)$ for large $|\omega|$. We can also see the role of the psuedo-random elements $R_{mn}$, albeit without the

---

[1]The Mathematica notebook including the codes and the results is available as an ancillary file at https://arxiv.org/abs/2212.03914.

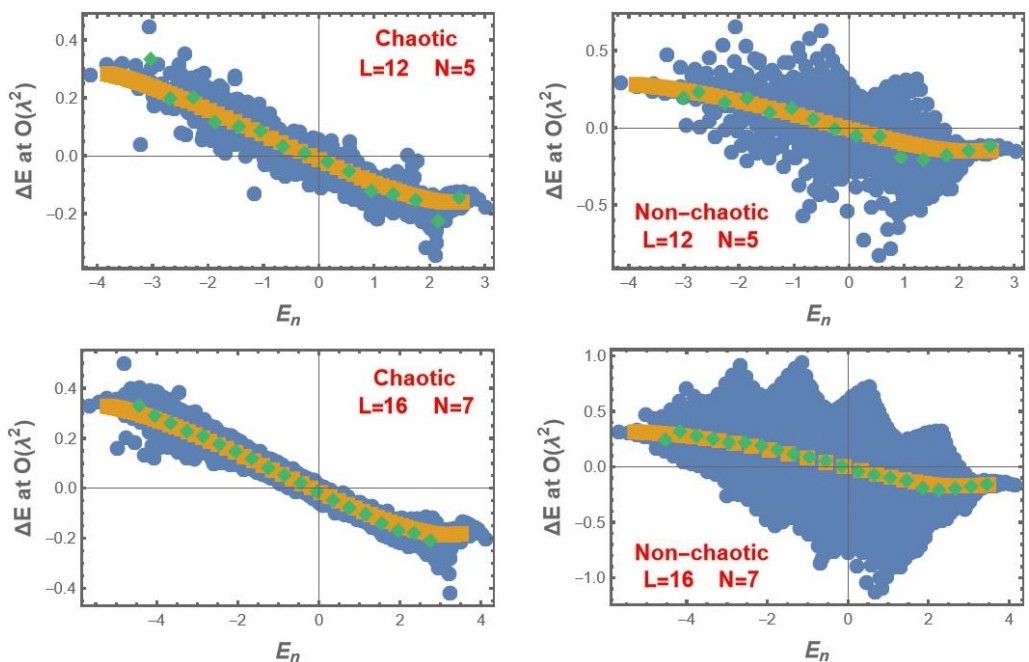

Figure 3: The leading term of the change in energy $\Delta E$ for chaotic and non-chaotic XXZ spin chains after a unit delta function perturbation $\lambda(t) = \delta(t)$. The blue dots are for the different energy eigenstates of energy $E_n$ as the initial state, the yellow plots are for thermal states with energy $E_n$ and the green dots are for microcanical ensembles.

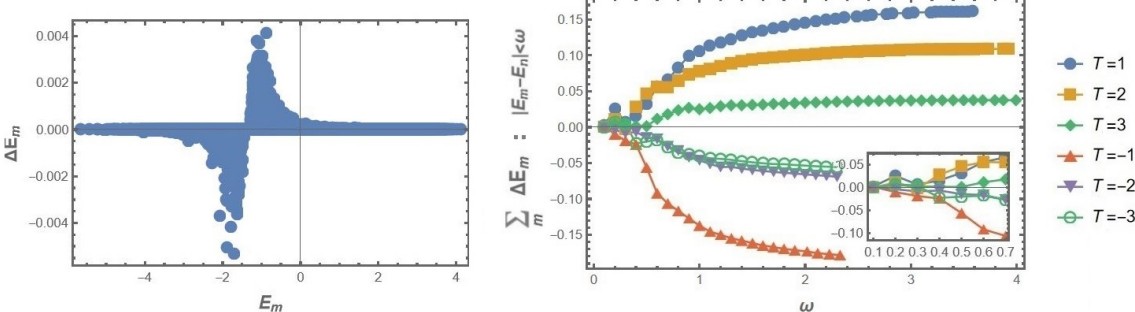

Figure 4: Left: $\Delta E_m$ as a function of $E_m$ for an energy eigenstate $E_n$ with effective temperature $T = 2$. Right: Sum of $\Delta E_m$ from $-\omega$ to $\omega$ as a function of $\omega$ for different energy eigenstates $|n\rangle$ with effective temperatures $T = 1, 2, 3, -1, -2, -3$. The inset shows that the sum does not grow for $\omega < 0.5$.

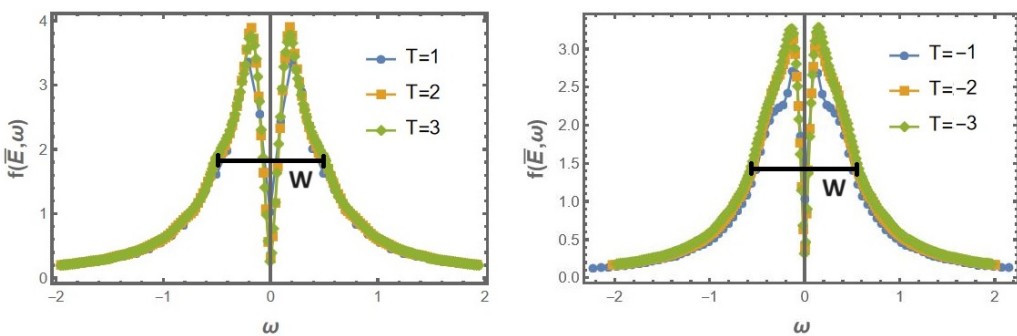

Figure 5: $f(\bar{E}, \omega)$ as a function of $\omega$ for different effective temperatures. The half of the bandwidth $W/2 \sim 0.5$.

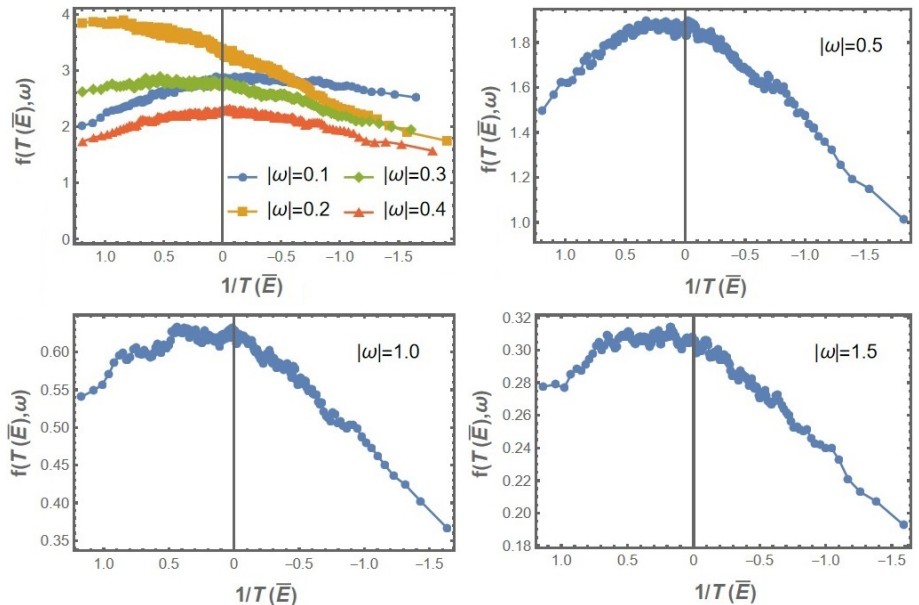

Figure 6: $f(\bar{E}, \omega)$ as a function of the effective inverse temperature $1/T(\bar{E})$ for different $|\omega|$. The observable is the total spin current in the chaotic XXZ spin chain.

signs. The right panel in Figure 4 is the plot of the sum of $\Delta E_m$ from $-\omega$ to $\omega$ as a function of $\omega$. It shows that the bulk of $\Delta E$ value comes from the region $0.5 \lesssim \omega$. We will find that the bandwidth $W/2 \sim 0.5$.

Figure 5 are the plots of $f(\bar{E}, \omega)$ as a function of $\omega$ for different values of $\bar{E}$ which are identified in terms of effective temperature. We can see that the bandwidth $W \sim 1$. As mentioned above in section 3, the physics inside this band is largely governed by RMT. Note that in this particular case, $f(\bar{E}, \omega) = 0$ at $\omega = 0$ from the symmetry of the model.

Figure 6 are plots of $f(\bar{E}, \omega)$ as a function of the effective inverse temperature of $\bar{E}$. These plots show that $f(\bar{E}, \omega)$ is indeed a monotonically increasing function of $S(\bar{E})$ or the magnitude of the effective temperature $|T|$ for most values of $\omega$. In all reasonable physical systems, the entropy is a monotonically increasing function of the magnitude of the temperature $|T|$. For $|\omega| = 0.2$, $f(\bar{E}, \omega)$ is not a monotonically increasing function. But this particular value of $\omega$ is within the band $\{-W/2, W/2\}$ where the physics is governed largely by RMT.

The edges in the energy spectrum of the XXZ spin chain are less chaotic [21]. This results in the unexpected behaviour at low temperature in the plots of $f(\bar{E}, \omega)$. To verify this, we also calculated $f(\bar{E}, \omega)$ in the highly chaotic ($q = 4$) SYK model [24, 25]. It is well known that even the edges of the spectrum are chaotic in SYK model [26]. We consider the occupation number at a single site as the observable so it corresponds to a local operator. It has been shown that this operator satisfy the ETH statement [27]. Figure 7 are the plots of $f(\bar{E}, \omega)$ as a function of the effective inverse temperature of $\bar{E}$ for different values of $|\omega|$ [2]. In this case, we find the expected behaviour even at very low temperature.

---

[2]The Mathematica notebook including the codes and the results is available as an ancillary file at https://arxiv.org/abs/2212.03914.

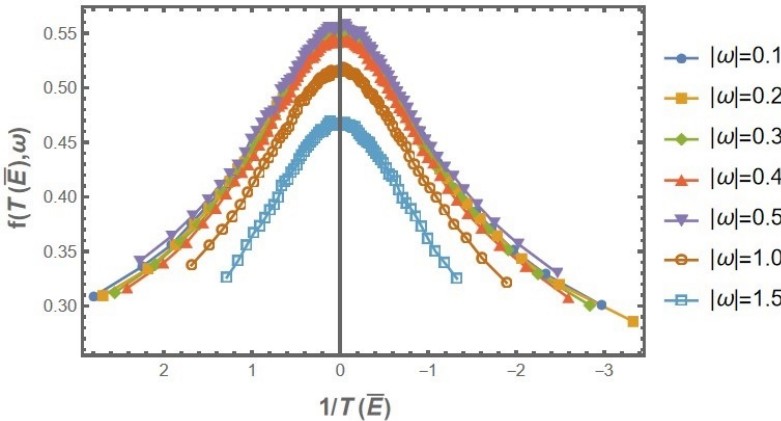

Figure 7: $f(\bar{E}, \omega)$ as a function of the effective inverse temperature $1/T(\bar{E})$ for different $|\omega|$. The observable is the occupation number at a single site in the chaotic ($q = 4$) SYK model. Number of site $L = 16$ and number of fermions $N = 7$.

## 5 Higher order terms of the change in energy

We consider the higher order terms of the perturbation theory in this section. The $O(\lambda^3)$ term of the change in energy $\Delta E$ is

$$\frac{i}{2}\sum_{l,m} E_l\left[\tilde{\lambda}(\omega_{nm})\tilde{\lambda}(\omega_{ml})\tilde{\lambda}(\omega_{ln})\mathcal{O}_{nm}\mathcal{O}_{ml}\mathcal{O}_{ln} - \tilde{\lambda}(\omega_{nl})\tilde{\lambda}(\omega_{lm})\tilde{\lambda}(\omega_{mn})\mathcal{O}_{nl}\mathcal{O}_{lm}\mathcal{O}_{mn}\right] \quad (11)$$

where $\omega_{nm} = E_n - E_m$. This term is identically zero because $\tilde{\lambda}(\omega) = \tilde{\lambda}(-\omega)^*$, $\mathcal{O}_{nm} = \mathcal{O}_{mn}^*$ and the sums run over the entire Fock space. Similarly, all odd power terms of $\lambda(t)$ are identically zero.

The $O(\lambda^4)$ term of the change in energy $\Delta E$ is

$$\frac{1}{4}\sum_{k,m,l}(E_n + E_l - E_k - E_m)[\tilde{\lambda}(\omega_{nk})\tilde{\lambda}(\omega_{kl})\tilde{\lambda}(\omega_{lm})\tilde{\lambda}(\omega_{mn})\mathcal{O}_{nk}\mathcal{O}_{kl}\mathcal{O}_{lm}\mathcal{O}_{mn}] \quad (12)$$

In general, this sum is negative (positive, zero) for initial states with positive (negative, infinite) temperature. $R_{mn}$ are not completely random [12]. If they were completely random, these higher order terms will be suppressed by exponentials of the entropy and a full analytic proof of non-negative $\Delta S$ is obvious.

In general, the even powers of $\lambda(t)$ have definite signs (statistically speaking) and the higher order terms are suppressed. Figure 8 are plots of $\lambda(t)^4$ and $\lambda(t)^6$ terms in $\Delta E$ series expansion for the chaotic XXZ spin chain ($L = 16, N = 7$) after a unit delta function perturbation $\lambda(t) = \delta(t)$. We can see that these higher order terms are suppressed compared to the leading order term even with a unit perturbation strength.

## 6 Conclusions

We show that quantum chaotic systems inherently possess an arrow of time. Working at leading order in perturbation theory, we show that the second law of thermodynamics constrains the off-diagonal terms in ETH statement. The new constraint has to be taken into account when one is working beyond probe limit. If the system is at a given energy $E$, the energy

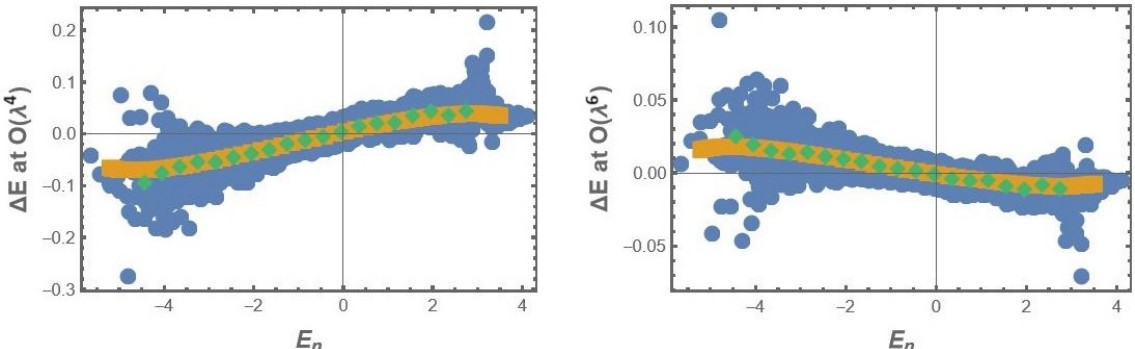

Figure 8: $\lambda(t)^4$ and $\lambda(t)^6$ terms in $\Delta E$ series expansion for the chaotic XXZ spin chain ($L = 16, N = 7$) after a unit delta function perturbation $\lambda(t) = \delta(t)$. The blue dots are for the different energy eigenstates of energy $E_n$ as the initial state, the yellow plots are for thermal states with energy $E_n$ and the green dots are for microcanical ensembles.

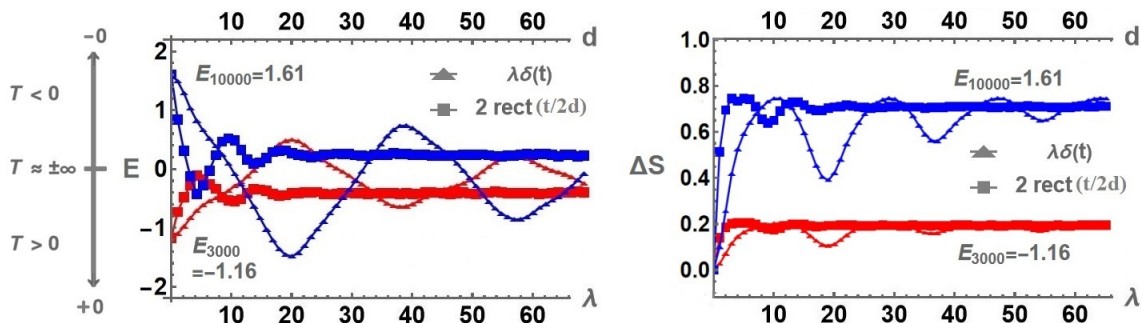

Figure 9: Left: Change in energy after large perturbations in the chaotic XXZ spin chain. Starting from the positive temperature state, the system always gains energy (red curves). Starting from the negative temperature state, the system always loses energy (blue curves). Right: The change in entropy is always postive.

eigenstates which are at a distance greater than $W/2$ in the energy spectrum play a very important role in the chaotic dynamics of the system. It has already been shown that these states constrain the Lyapunov exponent [11].

We also calculated the higher order terms of $\lambda(t)$ in $\Delta E$. For reasonable perturbation strength, the higher order terms are small compare to the leading term. The odd power terms of $\lambda(t)$ are identically zero.

We also study perturbations with large values of $\lambda(t)$ numerically. We considered perturbations of the form $\lambda(t) = \lambda \delta(t)$ and $\lambda(t) = 2\,\text{rect}(t/2d)$ which is the rectangular function of height 2 and width $2d$. $\Delta E$ and $\Delta S$ as a function of $\lambda$ and $d$ are plotted in Figure 9. $\Delta E$ remains always positive (negative) for positive (negative) initial temperature. This agrees with the Kelvin's form of the second law of thermodynamics. The change in entropy is also always positive in these examples. Note that the energy even jumps the infinite temperature region.

We can still reduced the entropy by using a fine-tuned perturbation. This can be accomplished by predominantly turning on modes $\tilde{\lambda}(\omega)$ which allows transition to only lower energy levels. But this fine-tuning requires the knowledge of a large part of the spectrum and precise knowledge of the initial state. This is reminiscent of the Maxwell's demon problem in classical thermodynamics. In the present case, finding the energy levels at the required precision involves performing highly precise measurements which increases the total entropy of the system and the outside agent.

# Acknowledgements

The author would like to thank Gautam Mandal for helpful discussions during one of which the main idea for this paper arose.

**Funding information**    The author is fully supported by the Department of Science and Technology (Government of India) under the INSPIRE Faculty fellowship scheme (Ref. No. DST/INSPIRE/04/2020/002105).

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
