# Peer review of "Quantum chaos and the arrow of time"

_SciPost Physics_

## Round 9 · Referee Report · Anonymous (Referee 1) · 2024-5-9

Report

This manuscript presents a fresh idea, as original as it is nontrivial. In short, it suggests that ETH constrains the matrix elements of relevant observables more than it was originally conjectured, if one requires that these matrix elements support the second law of thermodynamics.

More precisely, the manuscript suggests adding a requirement that off-diagonal large energy difference matrix elements of an observable compliant with the second law increase monotonically with entropy.

Numerical tests summoned to support of the above assertion produce convincing results.

Several suggestion for improvement.

  1. I would refrain from publishing an assertion that "The main idea of ETH comes from Berry’s conjecture." At the time, there were several interrelated currents of thought, with no obvious causal relationship between them. When lecturing on the subject, I, usually, use Feingold-Perez paper as a logical precursor of the ETH, but I never mean that Mark Srednicki should have cited them or, even, that he knew about the existence of this paper.

  2. It would make sense if the authors cite Feingold-Perez paper [M. Feingold and A. Peres, Distribution of matrix elements of chaotic systems, Phys. Rev. A 34 (1986), 591] .

  3. Authors assert several times that for ETH to be valid, the observables must be "non-fermionic." I don't think this is correct. True, the classical limit is difficult in the fermionic case, but ETH doesn't require an existence of a classical limit . The formula (1) does, but ETH is bigger than that. It can be reformulated in terms of a loss of memory of the initial conditions = "eigenstates of the same energy all look the same".

  4. Authors should say more about the width $W$. Indeed, it is observable-dependent and hence hard to define. But it will suffice to say that $W$ is comparable with the HWHM of f(\omega).

  5. I don't think that the function $f$ is "of the order of one, " if anything, that would be impossible on dimensional grounds (all the terms in (1) should be measured in the same units as $O$; but both the "density of states" exponent and the random variable $R$ are dimensionless.) For the observables I am familiar with, $f$ is of the order of $O$.

  6. Conclusion section would benefit from repeating the main assertion of the manuscript explicitly.

Recommendation

Publish (easily meets expectations and criteria for this Journal; among top 50%)

  • validity: high
  • significance: high
  • originality: top
  • clarity: top
  • formatting: perfect
  • grammar: perfect

Author:  Nilakash Sorokhaibam  on 2024-06-20  [id 4576]

(in reply to Report 1 on 2024-05-09)

Dear referee, Thank you for sending in the report quickly and portraying the correct historical context under which ETH was developed. Let us reply to the report here while we wait for the other referee reports. The revised manuscript will be submitted later after receiving the other reports.

I would refrain from publishing an assertion that "The main idea of ETH comes from Berry’s conjecture." At the time, there were several interrelated currents of thought, with no obvious causal relationship between them. When lecturing on the subject, I, usually, use Feingold-Perez paper as a logical precursor of the ETH, but I never mean that Mark Srednicki should have cited them or, even, that he knew about the existence of this paper.

It would make sense if the authors cite Feingold-Perez paper [M. Feingold and A. Peres, Distribution of matrix elements of chaotic systems, Phys. Rev. A 34 (1986), 591] .

Thank you for pointing out the Feingold-Peres paper which is an extension of an earlier Feingold-Moiseyev-Peres paper (Chem. Phys. Lett. 117, 344 (1985)) to off-diagonal matrix elements. This earlier paper cited Berry's paper prominently. We will remove the misleading sentence "The main idea of ETH comes from Berry’s conjecture" and we will cite the suggested Feingold-Peres paper as "...off-diagonal matrix elements of observables in energy eigenstates for a quantum chaotic system were studied in detail in [Feingold-Peres paper]".

Authors assert several times that for ETH to be valid, the observables must be "non-fermionic." I don't think this is correct. True, the classical limit is difficult in the fermionic case, but ETH doesn't require an existence of a classical limit . The formula (1) does, but ETH is bigger than that. It can be reformulated in terms of a loss of memory of the initial conditions = "eigenstates of the same energy all look the same".

One can construct highly non-trivial operators which can differentiate "eigenstates of the same energy". This is well known among black-hole-information experts (see page 63 in this report). These non-trivial observables also do not usually thermalize in a typical state even after long time evolution. ETH is useful and popular because it turns out that these non-trivial operators are very hard to measure or calculate. So, the observational perspective seems to be very important. Moreover, we can only perturb a system using a classical source which couples to a non-fermionic operator. All observables are non-fermionic. This is why we only consider non-fermionic operators.

Please note that we have not considered explicitly coupling the system with another system (or a bath). We only perturb the system using a time-dependent classical source. Indeed one can couple two systems using fermionic operators. For example, in this Maldacena-Qi paper, coupling of two identical SYK systems (L for left system and R for right system) is considered. The coupling term is of the type $i\mu\psi_L\psi_R$ where $\psi_L$ and $\psi_R$ are Majorana fermions of the L and R systems. But this does not mean that we can consider $(i\mu\psi_L)$ to be a classical source for the R-system because $\psi_L$ is a quantum operator. If we integrate out the L quantum degrees of freedom, considering a given state of the full (L+R) system, we would be left with the right system perturbed by a classical source coupled to a non-fermionic operator of the R-system.

Authors should say more about the width W. Indeed, it is observable-dependent and hence hard to define. But it will suffice to say that $W$ is comparable with the HWHM of $f(\omega)$.

We agree that we should have written more about the width $W$. Three scales of $\omega$ have been studied numerically (Figure 17 in Ref. No. [4]), viz., large, intermediate and small $\omega$. The small $\omega$ corresponds to the diffusive RMT behaviour and it scales as $\omega\sim L^{-2}$ where $L$ is the system size. The intermediate $\omega$ corresponds to ballistic dynamics and scales as $\omega \sim L^{-1}$. In both the intermediate $\omega$ and the small $\omega$, $f$ does not fall exponentially and scales roughly as $L^{1/2}$. So, the statement that $f$ falls exponentially for $|\omega|\gtrsim W/2$ is still correct where $W/2$ separates the intermediate $\omega$ and the large $\omega$. But inside $(-W/2,W/2)$, there is segregation of the intermediate $\omega$ and the small $\omega$. Hence, we realize that referring to RMT only is not the full picture. We plan to write that $W\sim L^{-1}$ and it is roughly taken to be HWHM of $f(\omega)$. We will drop the sentence "It is well known that the long time physics from the band |ω| < W/2 is governed by RMT."

I don't think that the function f is "of the order of one, " if anything, that would be impossible on dimensional grounds (all the terms in (1) should be measured in the same units as $O$; but both the "density of states" exponent and the random variable R are dimensionless.) For the observables I am familiar with, $f$ is of the order of $O$.

Thank you pointing this out. This is clearly an oversight. Indeed near the diagonal (small $\omega$), I can infer from the Feingold-Peres paper that f is of the order of $\mathcal{O}(\bar{E})$. So, "is of the order of 1" will be replaced with "is of the order of $\mathcal{O}(\bar{E})$".

Conclusion section would benefit from repeating the main assertion of the manuscript explicitly.

This suggestion will be incorporated in the revised manuscript.

We will also incorporate some other small changes and corrections in the revised manuscript. The details of the changes will be clearly documented in the formal reply.

regards, NS

---

## Round 9 · Referee Report · Anonymous (Referee 2) · 2024-6-20

Report

This paper claims to show that an arrow of time arises naturally in quantum chaotic systems. The argument begins with the claim that, in a many-body quantum chaotic system, a generic time-dependent perturbation (which occurs over a finite time) will cause a change in energy ΔE that has the same sign as the temperature T. The author then invokes the thermodynamic relation ΔE = T ΔS to argue that ΔS is always positive, and then invokes the second law of thermodynamics to argue that this shows the existence of an arrow of time.

In my view, this is a circular argument, because the arrow of time that is implied by the the second law is assumed rather than proved. Relatedly, the author does not provide a dynamic definition of entropy as a quantity that could be computed given the quantum state (pure or mixed). My opinion is that this is a fundamental flaw in the paper.

Furthermore, there is a technical error that negates the author's claim about the E dependence of f(E,ω). The "naive substitution of the ETH expression" invoked to go from eq.(6) to eq.(7) is not correct. The correct way to do this is given in ref.[11], and involves taking greater care with the various density-of-states factors that arise. When this is done, there is an additional factor of exp(βω/2) in the integrand of eq.(7), where β=1/Τ. [Note that there is also a typo in this equation: a factor of |λ(ω)|^2 is missing.] This factor will yield the sign relation between ΔE and T claimed by the author without requiring any properties of the E dependence of f(E,ω).

Because of these issues I cannot recommend publication.

Recommendation

Reject

  • validity: poor
  • significance: -
  • originality: -
  • clarity: -
  • formatting: -
  • grammar: -

Author:  Nilakash Sorokhaibam  on 2024-06-21  [id 4579]

(in reply to Report 2 on 2024-06-20)

(Reply to referee report 2, dated 20 June 2024)

Dear referee,

Thank you for the report. Following is our reply.

The referee writes:
"In my view, this is a circular argument, because the arrow of time that is implied by the the second law is assumed rather than proved."

Our response:
We do not assume second law or the arrow of time. Indeed, energy eigenstates of integrable systems inherently do not possess second law. So, *the crux of the argument is visible in Figure 3*. Using the thermodynamic relation, we could have as well plotted $\Delta S$ as a function of $E_n$. Then it would have been clear that energy eigenstates of chaotic system obey second law while energy eigenstates of integrable system do not obey second law. But there is benefit for studying in terms of $\Delta E$, meaning, in terms of Kelvin's statement of second law. Please note that we are consider a single energy eigenstate to be the initial state.

The main idea is to test whether change in energy $\Delta E$ also satisfy ETH or Berry's conjecture. We know that usual observables like occupation number, etc., obey ETH in a chaotic system, their expectation value in an energy eigenstate is very close to the thermal expectation value. But how about $\Delta E$ when we perturb the system starting from an energy eigenstate? It should also obey ETH otherwise as we wrote "one would be able to differentiate between an energy eigenstate and the corresponding thermal state by performing a small perturbation experiment and measuring the change in energy" which is against the idea of ETH or Berry's conjecture.

The referee writes:
"Relatedly, the author does not provide a dynamic definition of entropy as a quantity that could be computed given the quantum state (pure or mixed)."

Our response:
We are aware of multiple attempts to define dynamical entropy. But we do not need such novel definitions, we are only using the standard textbook definition of entropy.

Here is where the Kelvin's form of second law becomes useful. If the system starts with entropy $S$ and temperature $T$, after a small perturbation let us say the system gains energy $\Delta E$. Even though the system will take time to thermalize, the final entropy will be $S+\Delta E/T$.

The referee writes:
"The 'naive substitution of the ETH expression' invoked to go from eq.(6) to eq.(7) is not correct. The correct way to do this is given in ref.[11], and involves taking greater care with the various density-of-states factors that arise. When this is done, there is an additional factor of exp(βω/2) in the integrand of eq.(7), where β=1/Τ."

Our response:
There is no factor of exp(βω/2) when the initial state is a single energy eigenstate $|n\rangle$. Again, we are checking whether $\Delta E$ of single energy eigenstates obeys ETH or Berry's conjecture. So, the initial states under consideration are single energy eigenstates.

Ref.[11] performed the calculation in a thermal density matrix $\rho=e^{-\beta H}/Z$. It is known that $\Delta S\geq0$ starting from a thermal density matrix. We also mentioned it in the manuscript citing Ref. [9]. In fact, we are asserting and providing numerical evidences that single energy eigenstates should behave in the same manner *for a chaotic system but not for an integrable system* using ETH and Berry's conjecture.

The referee writes:
"Note that there is also a typo in this equation: a factor of |λ(ω)|^2 is missing."

Our response:
Thank you for pointing out this typo. We will correct it in the revised manuscript. This does not affect the central argument because $|\tilde{\lambda}(\omega)|^2$ is an even function of $\omega$.

We hope this reply addresses the points raised by the referee.

regards,
NS

---

## Round 9 · Referee Report · Anonymous (Referee 3) · 2024-7-12

Strengths

  1. The main claim is nontrivial and (if true) very important.
  2. The available numerical evidence clearly supports the claim.

Weaknesses

  1. The analytic part of the argument seems to contain a critical error, at least as currently written.
  2. The numerical evidence on its own is not enough to fully justify the claim (though I'll suggest how this could be changed).
  3. The presentation is often lacking in detail.
  4. Many background statements are insufficiently justified by argument or reference.
  5. The introductory parts read too much like popular science or PR ('public relations').

Report

The paper presents an analytic argument for the following: suppose we want an ETH-compliant system to also comply with the Second-Law requirement that after any perturbation of a thermalized state, the re-thermalized state must have higher entropy than the initial state. (Let's call this the 'perturbation increases entropy' requirement.) Then it is necessary that the function $f(E, \omega)$ (which appears in the ETH) satisfy an additional condition: for every fixed $\omega$ (except possibly those within a certain interval centered at zero), $f(E, \omega)$ must be a monotonically increasing function of $S(E)$. The author then verifies numerically that $f(E, \omega)$ indeed has this property in two quantum-chaotic systems, which are also shown to comply with the 'perturbation increases entropy' requirement. Also shown numerically is that an integrable system violates this requirement.

If all this is really correct---and the numerics, at least, suggest that it is---then I believe that this could turn out to be an important paper. (I hope this belief comes some way toward justifying the length of this report, for which I apologize.)

But some major changes will have to be made first. In particular, I agree with Referee 2 that the analytical part of the argument (Sec. 3 of the paper) contains a critical error, at least as currently written. The paper cannot be published unless this can be fixed, either analytically or numerically.

My explanation of this error is in Remark 1, below. It makes no use of the thermal density matrix, so the author's reply to Referee 2 does not address my formulation.

If it turns out that the author is unable to fix the analytical argument, then I think the current numerical evidence is not enough to carry the paper on its own..

However, I also think that the author may be able, using the numerical codes he already has, to relatively quickly produce additional numerical evidence, which may be enough to replace the analytic argument. What is needed is numerical evidence showing that if, in an ETH-compliant system, the function f(E, omega) does not satisfy the additional condition, then the system will violate the 'perturbation increases entropy' requirement.

It is important that the violation occur despite the fact that the standard ETH holds. Otherwise, we leave open the possibility that the standard ETH, even without the additional condition on $f(E, \omega)$, may be enough to ensure compliance with the 'perturbation increases entropy' requirement. This is why the paper's results on the integrable system don't help here.

In my Remark 2 below, I state more explicitly how to do this computation.
* * *
Before proceeding to my remarks about the paper, let me address Referee 2's other two objections.

One of these objections is that the paper employs a circular argument---that is assumes the Second Law to prove the Second Law. To this, I reply that if one removes from the paper (as I will urge the author to do) all of the PR-sounding text (PR as in 'public relations')---in particular, all mentions of 'the arrow of time'---then the substance of the paper's claims and arguments is clearly non-circular and non-trivial. Indeed, this substance is summarized in the very first paragraph of this report, and I believe it is manifestly non-circular. See Remark 29, below, for further details.

The other objection is that the paper doesn't provide 'a dynamic definition of entropy as a quantity that could be computed given the quantum state (pure or mixed)'. My reply is again that one should look at the 'cleaned-up' version of the paper's claim and argument (as in the first paragraph of this report). That version clearly stands on its own, even without introducing the kind of entropy asked for by Referee 2.

Having said that, there already exists a notable proposal for just such a definition (the so-called 'd-entropy'; Polkovnikov, Ann. Phys. 326, 486 (2011); Santos, Polkovnikov, and Rigol, Phys. Rev. Lett. 107, 040601 (2011)). And yes, it would be interesting to see what happens to the d-entropy during the processes studied in the present paper. But I don't think the present paper is incomplete without such a study.

In this regard, I would like to caution anyone reading this from falling into the following trap: adopting the 'tacit understanding' that the Second Law 'implies a continuous increase in some property called entropy, which was supposedly defined for systems out of equilibrium' (Lieb and Yngvason, Phys. Today 53, 32, 2000). I suspect this trap activates upon one's encounter with Boltzmann's H-theorem. Lieb and Yngvason in fact argue that for systems not in equilibrium, 'it is generally not possible to find a unique entropy that has all relevant physical properties' (Proc. R. Soc. A 469, 20130408 (2013)). But regardless of whether they are correct, it is indisputable that the standard, equilibrium entropy is enough to answer questions about any physical processes whose initial and final states are equilibrium states. There is no restriction on the intermediate states, which may be arbitrarily far from equilibrium. About such processes, the Second Law says that if there is no heat transfer involved, then the entropy of the final state must be greater than the entropy of the initial state. (Here the notion of 'no heat transfer involved' should be replaced by the notion of 'adiabaticity'. This latter notion can be defined without invoking the notion of heat; see p. 17 of Lieb and Yngvason, Phys. Rep. 310, 1 (1999))
* * *
In what follows, if a function $f(E, \omega)$ is a monotonically increasing function of $S(E)$ for all $\omega$ outside a certain appropriate window, I will say that $f(E, \omega)$ is 'ETH-monotonic'.

I would urge the author to either adopt this shorter terminology, or else come up with a different, but similarly short, terminology of his own.

***** ********************
Remark 1. (major concern: we don't need ETH-monotonicity because the density of states does the job)
* * *
Here is an argument that seems to show that one gets the expected sign of delta E even if f(E, omega) is not ETH-monotonic.

Let's look at Eq. (6). We want to convert the sum to an integral. We keep $E_{n}$ fixed, promote $E_{m}$ to a continuous variable $E$, and replace the sum over $m$ with an integral over $E$ (for simplicity it can be from -infinity to +infinity). In going from a sum to an integral, we need to include an appropriate measure, namely, $e^{S(E)}$). That is, we have the replacement $\sum_{m}\rightarrow \int_{-\infty}^{\infty} dE\, e^{S(E)}$. (This is as in Ref. 11, between Eqs. (6) and (7) of that paper.) Next, we change the integration variable to $\omega = E - E_n$. We get the following: $\Delta E = \int_{-\infty}^{\infty} dE\,\omega\, |\tilde{\lambda}(\omega)|^2 \,[f(E_n +\omega/2, \,\omega)]^2 e^{S(E_n+\omega)-S(E_n+\omega/2)}$ . Note the presence of the density-of-states terms, $e^{S(E_n+\omega)-S(E_n+\omega/2)}$. In particular, they don't cancel, and we now need to understand what they will do to the integral. Let's look at how the exponent behaves as a function of $\omega$ for a fixed $E_n$. So, let $s(\omega)=S(E_n+\omega)-S(E_n+\omega/2)$.

Let's assume that $S(E)$ is concave down. (That is, $S''(E)<0$ for all $E$. Recall that this is actually a general requirement on the entropy function; see e.g. Callen's textbook, https://tinyurl.com/2jk2r3ca). Let's also assume $S(E)$ has a maximum at $E=0$, as in Fig. 1. Here $S(E)$ may or may not be an even function. (As a simple example, in Fig. 1, it seems $S(E)$ could approximately be of the form $S(E) = -(E/a)^2+b$ , resulting in the Gaussian-looking curve seen in the plot.) Equivalently, we assume that negative energy states have positive temperatures, while positive energy states have negative temperatures.

In that case, one can show that the exponent $s(\omega)$ has two zeroes: one at 0, and one for some $\omega$ between $-E_n$ and $-2 E_n$. Moreover, $s(\omega)$ is positive only for $\omega$'s that lie between the two zeros.

The easiest way to see this is simply to simply sketch it (e.g. see this anonymously shared image on Google Docs: https://tinyurl.com/2p8kbm8d ). It is not hard to do a formal proof, but this review will already be too long. As a simple example, if $S(E) = -(E/a)^2+b$, then $s(\omega)=- \frac{3}{4 a^2} \,\omega\, (\omega + 4 E_n/3)$.

Therefore, if $E_n$ is negative, so that $-E_n$ and $-2 E_n$ are positive, then $s(\omega)$ is positive for all $\omega$'s between 0 and some positive $E^*$, where $ 0 < |E_n| < E^* < 2|E_n|$. So $e^{s(\omega)}$ has a sharp peak for some positive $\omega^*$, where $0 < \omega^* < E^*$.

As a consequence, in our integral for $\Delta E$, the positive values of omega will be enhanced, which will tend to make our integral for $\Delta E$ be positive, as the Second Law says it should be (given that $E_n$ is negative, corresponding to positive temperature). Note that it is part of standard ETH that the magnitude of the off-diagonal matrix elements is set by $e^{-S(E)}$ (as in Eq. (1)). Thus, even if $f(E, \omega)$ is not ETH-monotonic, it should nevertheless be incapable of overpowering the sharp peak due to $e^{s(\omega)}$. So $\Delta E$ will have the correct sign simply due to that sharp peak.

The argument for positive $E_n$ is similar, except that now $e^{s(\omega)}$ has a sharp peak at some negative value of $\omega$, which tends to make the integral (and thus $\Delta E$) negative; which, of course, is just what the Second Law says it should be for positive $E_n$, corresponding to negative temperature.

So it seems that, assuming just the standard ETH, the density-of-states terms in the integral are enough, all by themselves, to ensure that the sign of$\Delta E$ agrees with the prediction of the Second Law. We don't need to assume any special behavior of $f(E, \omega)$.

Incidentally, to make contact with what Referee 2 said, let's for the moment consider the case when $|\omega| \ll 1$. Then we have that $s(\omega)=S(E_n+\omega)-S(E_n+\omega/2) = \frac{1}{2} S'(E+\omega) \, \omega + \mathcal{O}(\omega^2)$. Using the thermodynamic relation $\frac{d S}{d E} = \frac{1}{T} = \beta$, we get $s(\omega)= \frac{1}{2} \,\beta\, \omega + \mathcal{O}(\omega^2)$; thus we get the factor $e^{\beta \omega/2}$ mentioned by Referee 2.
Of course, since our integral over omega goes from -infinity to +infinity, we can't actually assume that $|\omega| \ll 1$. But it is nevertheless interesting that the factor $e^{\beta \omega/2}$appears here.
* * *
I actually hope that the argument I just presented is wrong. After all, numerical evidence suggests that $f(E, \omega)$ does have the behavior postulated by the author, and one would think there should be a good reason for that. Perhaps the density-of-states peak is actually not sharp enough to produce the effect on its own? In any event, further argument is needed to analytically justify the author's claim that $f(E, \omega)$ must be ETH-monotonic.

Requested changes

Warnings issued while processing user-supplied markup:

  • Inconsistency: plain/Markdown and reStructuredText syntaxes are mixed. Markdown will be used.
    Add "#coerce:reST" or "#coerce:plain" as the first line of your text to force reStructuredText or no markup.
    You may also contact the helpdesk if the formatting is incorrect and you are unable to edit your text.

* Remark 2 (how numerics alone could be made convincing) *

Even if the analytical argument doesn't go through, the main claim [that f(E, omega) must be ETH-monotonic] could still be correct.

Note that the author has already shown numerically that perturbing the XXZ chaotic system always produces a $\Delta E$ with the expected sign, while perturbing an integrable system often doesn't. Moreover, the author has shown numerically that, for the XXZ chain and for another chaotic system, $f(E, \omega)$ is in fact ETH-monotonic. This is good as far as it goes, but it is incomplete. To complete the numerical case for the necessity of the ETH-monotonicity, should show that in a chaotic system in which ETH holds but ETH-monotonicity does not, we often get a $\Delta E$ with the wrong sign.

This could be done as follows. For the next-to-nearest neighbor XXZ chain, with the global spin-current as the operator of interest, the author has already computed all the matrix elements and extracted $f(E, \omega)$. Presumably, the author then also knows what the corresponding $R_{mn}$ and $e^{-S(\bar{E})/2}$ values are (this refers to Eq. (1)). Therefore, the author can use Eq. (1) to construct 'artificial' matrix elements which are just like the actual ones, except that the author will change their $f(E, \omega)$ by hand so that it is not ETH-monotonic. For example, one could flip the dependence of $f(E, \omega)$ on $E$, so that the artificial $f(E, \omega)$ becomes a monotonically decreasing function of $S(E)$ (for all $\omega$ outside a certain window). Then, using these artificial matrix elements, the author can recompute the leading term of delta E. If it comes up with the wrong sign, that will show that the ETH-monotonicity is really essential. If it doesn't---well, then the main claim of the paper is wrong.

In addition to 'flipping' $f(E, \omega)$, the author could also replace it by a constant average value. Or by the average value plus a random scatter of variance comparable to the variance if the actual $f(E, \omega)$.

Moreover, since the matrix elements of an operator completely define the operator, the author should also be able to study large perturbations for this artificial operator.

If my objection in Remark 1 is correct and the analytical argument fails, then I think the author really must do this computation with 'artificial' matrix elements before I could recommend that the paper be published

On the other hand, if the analytical argument actually does go through despite my objection in Remark 1, then I could recommend publication even without the computation with the 'artificial' matrix elements. However, I would urge the author to do that computation in any case. It will make the paper much stronger.

* other remarks on the substance of the paper **

* Remark 3 *

The paper says, 'So, we will not be concerned with minute differences between different ensembles when working with finite systems or other controversial cases' (p. 3). But will the differences between different ensembles really be minute? After all, the state space of the system has no more than ${16 \choose 7} = 11440$ dimensions, which is significantly less than the $300\,000$ the author says one usually needs. In Ref. 18, Figure 1, the difference between the microcanonical and the canonical ensembles is clearly visible, and the dimensionality here was ${21\choose 5} = 20\,349$. The authors comment that 'This is an indication of the relevance of finite-size effects, which may be the origin of some of the apparent deviations from thermodynamics seen in the recent numerical studies of refs 4 and 5.'

Perhaps the author is correct and in this paper, the differences are indeed minute. But the author should provide some evidence for that from the numerics. For example, both the entropy and the temperature could be computed from the density of states in the microcanonical ensemble, and compared to the result the author obtained using the canonical ensemble of the same energy as the pure state.

* Remark 4 *

How do we know that the perturbation (parametrized by $\lambda(t)$) counts as a process involving no heat transfer? This is important because unless the perturbation counts as such a process, we cannot deduce that the change in entropy must be positive. (After all, the transfer of heat can lower the entropy of a system.) Perhaps all that's needed here is a reference to some work on quantum thermodynamics.

* Remark 5 *

As I said in Remark 1, the analytic derivation of Section 3 should be, well, an actual derivation rather than a sketch of one, especially since there is some doubt (on the part of both Referee 2 and myself) on whether it is correct. Since there are no constraints of space, so no justification for not including it.

A good start (which indicates the level of detail I advocate for) would be to begin with the actual Dyson series and actually derive Eq. (6) from it (as obvious as that derivation may seem to the author, let's keep in mind that not everyone is looking at the Dyson series every day).

* Remark 6 *

The paper says, 'This naive substitution is routinely used to calculate correlation functions and study chaos and non-equilibrium dynamics (for example see [10]).' (p. 5). So does that mean that the results in [10] are wrong? (Are the results also wrong in other papers where the 'naive approach' is used?) If those results are nevertheless correct, how come? What enables the authors of those papers to get away with the 'naive approach'?

* Remark 7 *

The paper says, 'For a non-fermionic operator, the above expression is positive (negative or zero) in a thermal state of positive (negative or infinite) temperature' (p. 5). This statement should be proven or at least a reference given where it is proven.

* Remark 8 *

Is the expression in Eq. (10) correct? Shouldn't it be 'xy-yx' rather than 'xx-yy'? That is to say, shouldn't it be $\sum_{i} \left(S_{i}^{x} S_{i+1}^{y} -S_{i}^{y} S_{i+1}^{x} \right)$? (Or, which is the same, $2i\sum_{i} \left(S_{i}^{+} S_{i+1}^{-} -S_{i}^{-} S_{i+1}^{+} \right)$, where $S_{i}^{\pm} = (1/2) (\left(S_{i}^{x} \pm i S_{i}^{y}\right)$. See e.g. Prosen, Nucl. Phys. B 886, 1177–1198 (2014), the text right before Eq. (1) and Eq. (70); also see Thingna and Wang, Europhys. Lett. 104, 37006 (2013), Eq. (3).

* Remark 9 *

The paper says, 'in this particular case, $f(E, \omega) = 0$ at $\omega = 0$ from the symmetry of the model.' The paper should explain which symmetry that is and why it implies that $f(E, \omega=0) = 0$ .

* Remark 10 *

The paper says, '$R_{mn}$ are not completely random [12]. If they were completely random, these higher order terms will be suppressed by exponentials of the entropy' (p. 9).

First of all, normally, exponential suppression is difficult to overcome. The author should provide some indication of how it is even possible that a slight non-randomness of $R_{mn}$ could be enough to overcome the exponential suppression.

Moreover, when the sum is converted to an integral, we also get the $e^{S(E)}$ term from the measure, as in Remark 1. In that remark, I argued that various density-of-states terms were incorrectly handled in Sec. 3. Is the same error present here? Even if we don't pass to an integral, there will be a corresponding effect for the sum.

* other remarks on physical content of the paper **

* Remark 11 *

The paper says 'These results are familiar in the realm of classical physics. But our physical world is governed by quantum mechanics at microscopic level and so far a complete understanding of the quantum version of the second law is lacking' (p. 2). This suggests that there are aspects of the second law that are clear in classical mechanics but unclear in quantum mechanics. Before the discovery of the ETH, I would agree. But I'm not so sure this is correct once the ETH is accepted. In particular, what, precisely, is missing in order to have a 'a complete understanding of the quantum version of the second law'? Whatever this may be, is it really the case that there isn't a corresponding problem in classical mechanics? At the very least, a reference for the claim should be given.

Whatever changes are made here, corresponding chnages should be made in the abstract.

* Remark 12 *

The paper says, 'Traditionally, quantum chaos is identified from the study of energy level statistics'. One should add, 'and of delocalization measures for the complexity of eigenstates.' For the latter, one can cite Santos and Rigol, Phys. Rev. E 81, 036206 (2010).

* Remark 13 *

The paper says, 'Inevitably, the quantum operators corresponding to these observables are hermitian and non-fermionic in nature.' The author should explain why the non-fermionc nature is important, and/or give a reference.

* Remark 14 *

In the text following Eq. (1), the author introduces the symbol $W$ without defining it. Given that there is no problem with space (this is a regular article, not a letter), the author should define $W$ (as is done in Ref. 10, Eq. (4.5)).

* Remark 15 *

The paper says, 'The first manifestation is thermalization of non-equilibrium excited states which are typical' (p. 3). Different people have different things in mind when they talk about 'typical states'. The author should explain what meaning of 'typical' he has in mind, especially since the literature has many, many discussions of 'typicality' in quantum mechanics (e.g. Popescu, Short and Winter, Nature Phys. 2, 754 (2006); Goldstein et al., Phys. Rev. Lett. 96, 050403 (2006); Reimann Phys. Rev. Lett. 99, 160404 (2007)). (Basically, the author needs to repeat the appropriate parts of Ref. 10, presumably those in the paragraph following Eq. (4.8).)

* Remark 16 *

The paper says, 'One subtlety when preparing states using quantum quenches is that even integrable systems thermalize to generalized Gibbs ensembles (GGEs) in the large system size limit' (p. 3). I don't think it is standard to describe this behavior of integrable systems as 'thermalization'. They 'equilibrate', but don't thermalize. Reference 18 is very explicit about this: 'However, if the system possesses further conserved quantities that are functionally independent of the hamiltonian and each other, then time evolution is confined to a highly restricted hypersurface of the energy manifold. Hence, microcanonical predictions fail and the system does not thermalize.'

Whatever changes are made here, the corresponding changes should be made in the following phrase in the abstract: '…thermalize to a generalized Gibbs ensemble'.

* Remark 17 *

The paper says, 'The non-equilibrium states are generalized typical states taking into account all the other conserved charges of the integrable system. This result simply follows from central limit theorem and the local nature of the Hamiltonians [18]' (p. 3). As best as I can tell, Ref. 18 says nothing about the central limit theorem, and the local nature of the Hamiltonian is only mentioned in the Supplementary Discussion, in the context of justifying the assumption that the width of the energy distribution be small.

An appropriate reference should be given here.

* Remark 18 *

The paper says, 'So in this manner, finite perturbations or bump quenches bring out the stark contrast between non-equilibrium dynamics of quantum chaotic systems and quantum integrable systems' (p. 3). As best as I can tell, this has only been demonstarted for free scalar field with time-dependent mass, and the author should use more tenative language (e.g., 'finite perturbations or bump quenches have a potential to bring out…').

* Remark 19 *

(This is also related to Remark 4.)

The paper says, 'Our result in terms of the change in energy is the Kelvin’s form of the second law of thermodynamics [20]' (p. 4). This would seem to need a bit of explaining.

After all, according to Huang's Statistical Mechanics, Kelvin's formulation is 'There exists no thermodynamic transformation whose sole effect is to extract a quantity of heat from a given heat reservoir and to convert it entirely into work.' On the other hand, the present paper considers a time-dependent perturbation of an initially thermalized state and asserts that the change in energy (upon re-thermalization) must have the same sign as the temperature. What counts as 'heat' and what as 'work' in this process? The relationship to Kelvin's statement is by no means immediate.

I would imagine that the best course of action here is to just find a paper that asserts that this is what the Second Law comes down to in such cases (namely, in cases of time-dependent perturbation of an otherwise isolated finite quantum system), cite it, and move on.

* Remark 20 *

There should be a reference for Eq. (8) and for the expression of the retarded Green's function.

* Remark 21 *

The paper says, 'Invoking Berry’s conjecture and ETH, we infer that the leading term in (6) and the thermal value should match. (p. 5). Why do we here need to invoke Berry's conjecture separately from the ETH?

* Remark 22 *

The paper says, '$f(E, \omega)$ is a constant function in random matrix theory (RMT)' (p. 5). A reference for this is needed.

* Remark 23 *

The paper says, 'It is well known that the long time physics from the band $|\omega| < W/2$ is governed by RMT' (p. 5). Again, a reference for this claim is needed.

* Remark 24 *

The paper says, 'This results in the unexpected behaviour at low temperature in the plots of $f(E, \omega)$' (p. 8). The author should spell out what 'unexpected behavior' he has in mind. For example, is he talking about the slight 'bump' at $1/T=-1.03$ in the $|\omega|=1.5$ plot, and the slight 'dip' in the same plot at $1/T=0.97$?

* Remark 25 *

The paper says, 'In general, this sum is negative (positive, zero) for initial states with positive (negative, in- finite) temperature' (p. 9). Is this known analytically or numerically? If analytically, a derivation should be given.

* Remark 26 *

In the study of large perturbations, how long was the relaxation time? Are we sure it is long enough for thermalization? It may be useful to include a sample time-dependent plot.

** remarks on presentation **

* Remark 27 *

The discussion of the numerical study of large perturbations should be in the main text, not in the Conclusions section.

* Remark 28 *

In several places, the paper talks of 'Fock space'. Normally, when Fock space is mentioned, most physicists expect to see creation and annihilation operators and occupation numbers. But I don't think any of that appears in this paper. It seems to me that what the author really means to say is 'state space' (as in here: https://en.wikipedia.org/wiki/Quantum_state_space); or, perhaps, 'Hilbert space', which in physics has come to denote both finite- and infinite-dimensional state spaces. Given that the numerics in this paper are (of course) done on finite-dimensional spaces, the term 'state space' seems to be the most appropriate.

* Remark 29 (How to respond to the charge of circular reasoning) *

Referee 2 brought up the interesting charge that the paper engages in circular reasoning, namely, that it uses the Second Law in one form to prove it in another form.

The author replied, in effect, that if the argument were truly circular, then the integrable system would also show the Second-Law-compliant sign of $\Delta E$, and it doesn't.

I don't think this particular reply is satisfactory. A circular argument for a proposition P need not be a simple tautology 'P, therefore, P'. It could instead start with 'P and Q', and then use a complicated chain of reasoning to formally prove P. This is still circular reasoning. But note that it is perfectly possible to start with 'P and Q' and then formally prove not-P! Indeed, this is often exactly what we do when we prove not-Q using a reductio ad absurdum argument. So, our situation may be as follows. Assuming the Second Law, in the case of the chaotic system we also recover the Second Law back. On the other hand, also assuming the Second Law, in the case of the integrable system we recover a negation of the Second Law. What this shows is that there are hidden additional assumptions that are false in the case of the integrable system. And if these hidden assumptions are true in the case of the chaotic system, then the reasoning for the chaotic system is indeed circular.

However, I believe this particular discussion is a red herring. To become immune to the circularity charge, all the author has to do is the following:

  1. He should remove all phrasing that seems to imply that the paper is attempting to 'derive' the Second Law, especially this one: 'In this work, we show that an arrow of time arises naturally in an isolated quantum system'. See also Remark 45.

  2. Early in the paper, he should point out that among the standard manifestations of the Second Law that apply to an isolated quantum system (that is, isolated except for a temporary perturbation), we have in particular the following two: A. $\Delta E = T \Delta S$; and B. following a perturbation, $\Delta S> 0$. The main result of the paper is this: suppose we have a system in which (i) the usual ETH holds, (ii) the entropy is a monotonically increasing function of $|T|$, and (iii), $\Delta E = T \Delta S$. Then, a necessary and sufficient condition for B. to hold in that system is for $f(E, \omega)$ to be ETH-monotonic.

Stated this way, there can be no charge of circularity. (Perhaps the author can devise an even better way to do this, but this way will certainly work.) The key is to refer to particular precise and quantitative manifestations of the Second Law, rather than simply to 'the Second Law'.

After all, this paper is studying systems which probe the limits of applicability of thermodynamics. It is entirely possible that various traditional 'manifestations' of the Second Law decouple in such edge cases: i.e., that such systems might comply with some of the 'manifestations' but not others. If one of our long-term ambitions is indeed to derive the Second Law from something more fundamental, then we need to understand under what circumstances its various 'manifestations' hold or not hold. So the work done in this paper is a very relevant contribution to this program. (See also the last paragraph of Remark 45.)

* Remark 30 *

The paper says, 'This suggests that $f (\bar{E}, \omega)$ is a monotonically increasing function of $S(\bar{E})$' (p. 5). I suppose I do understand what this is trying to say, but perhaps a bit of an explanation might be helpful. What could be confusing is that, for a fixed $\omega$, $f(E, \omega)$ is literally a function $E$, not of $S(E)$. So the author should add a clarifying note, perhaps something like: 'In other words, for most values of $\omega$, all of the following have the same sign: $\frac{\partial f}{\partial E}$, $S'(E)$, and $\frac{d|T|}{dE}$.'

(After all, the author is effectively defining $\tilde{f}(\tilde{S})=f(S^{-1}(\tilde{S}))$, where $\tilde{S}=S(E)$, and saying that $\frac{ f(S^{-1}(S))}{d S} > 0$ (it could be zero at isolated points). But $\frac{ f(S^{-1}(S))}{d S} =f'(E) (S^{-1})'(S(E)) = f'(E)/ S'(E) $, so $f'(E)/ S'(E) >0$, so $f'(E)$ and $S'(E)$ must have the same sign.)

* Remark 31 *

The paper says, 'To summarize, our two analytic results are 1. The change in energy at leading order… 2. $f(E,\omega)$ is a monotonically…' (p. 5). I wouldn't call 1. an 'analytic result'. It is an observation and a surmise, whose main purpose is to derive 2. It seems to me that only 2. counts as an 'analytic result'. (And that's certainly good enough—if Remark 1 can be answered.)

* Remark 32 *

The paper says, 'We will work with $L = 16$, $N = 7$ for the rest of the numerical results.' The paper should mention that the dimensionality of the state space in this case is ${16\choose 7} = 11440$.

* Remark 33 *

In Figure 5, both panels should have the same vertical scale.

* Remark 34 *

The paper says, 'We can also see the role of the pseudo-random elements $R_{mn}$, albeit without the signs' (pp. 7-8). Namely, what is it that we see? The 'bumpiness'?

* Remark 35 *

The paper says, 'Figure 5 are the plots of $f(E, \omega)$ as a function of $\omega$ for different values of $E$ which are identified in terms of effective temperature' (p. 8). How, exactly, was $f(E, \omega)$ computed? This should be explained in the text. (True, the Mathematica notebooks are attached. But those are hard to read, and anyway the paper should be self-contained.)

* Remark 36 *

The paper says, 'As mentioned above in section 3, the physics inside this band is largely governed by RMT' (p. 8). What implication does this have for the behavior of $f(E, \omega)$? Is there some feature in the numerically computed values of f(E, omega) that makes this RMT physics evident?

* Remark 37 *

The paper says, 'For $|\omega| = 0.2$, $f(E, \omega)$ is not a monotonically increasing function.'

I suppose this is referring to the fact that the slope of the orange cloud of points is negative just to the left of the y-axis. But if this is so, then I would say that $|\omega| = 0.1$ and $|\omega| = 0.3$ have similar issues: the slope of the green cloud is also negative just to the left of the y-axis, though the effect is smaller than for the orange points. And the blue cloud has a slighly positive slope to the right of the y-axis.

Of course, all these problematic values of omega are still within the band $[-W/2, W/2]$. But why highlight some of them and not others?

* Remark 38 *

In the sentence after 'This results in the unexpected behaviour at low temperature in the plots of $f(E, \omega)$. To verify this, we also …' (p. 8). Here 'this' (in 'verify this') doesn't fit. Presumably, the author wants to say, 'To verify that $f(E, \omega)$ has the expected behavior when the system is sufficiently chaotic, we also …'.

* Remark 39 *

In the next sentence, the paper says 'It is well known that even the edges of the spectrum are chaotic in SYK model [26]'. Where exactly is this stated in Ref. 26? I don't see it.

* Remark 40 *

The paper says, 'We can see that these higher order terms are suppressed compared to the leading order term even with a unit perturbation strength' (p. 10). When we compare the left panel of Fig. 8 with the bottom-left panel of Fig. 3, we see that in Fig. 3, the typical size is about 0.2, while in Fig. 8, for negative $E_n$, it is on the order of $0.1$ in magnitude. Most people would say that $0.1$ and $0.2$ are 'of the same order', and so they wouldn't say that one of them is 'suppressed' compared to the other. So instead of saying 'suppressed', at least for the quartic terms, one should instead say they are typically 'smaller by a factor of 2' (and that is enough to make the main point).

* Remark 41 *

The paper says, 'The new constraint has to be taken into account when one is working beyond probe limit' (p. 9). The author should explain what the 'probe limit' is, or at least give a reference where this is explained.

* Remark 42 *

The paper says, '… the dimension of the Fock space is only 792.' It would be good to say 'is only ${12\choose 5} = 792$.

* Remark 43 *

In the caption for Figure 5, add 'Left panel: positive temperatures. Right panel: negative temperatures.'

** remarks on the introductory parts *

* Remark 44 *

The paper says, 'A closer examination led us to a new profound constraint on the ETH statement' (p. 2). If other people decide this constraint is remarkable enough to be called 'profound', they can say so in their papers (with due credit to the author of the present paper). However, in this sentence, I strongly suggest that the word 'profound' be dropped. (But if the author insists on keeping it, that will not be a reason for me to not recommend publication. It really is just a suggestion, albeit a strong one.)

Whatever changes are made here, the corresponding changes should be made in the abstract.

* Remark 45 *

We should avoid hopelessly vague statements like 'In this work, we show that an arrow of time arises naturally in an isolated quantum system.' This may be appropriate for a popular science piece, but not for a technical research article.

Moreover, this statement can easily be interpreted (and apparently has been interpreted by Referee 2) as suggesting that the existence of the 'arrow of time' (which presumably is a stylistic variant of 'the Second Law of thermodynamics') will be in some way 'proven' in the article.

The article does no such thing. To prove or derive the Second Law, for example as formalized by Lieb and Yngvason (Phys. Rep. 310, 1, 1999), would entail deriving (from some other fundamental principles) each one of their 15 axioms. This 'might seem like a lot', they say on p. 5, but 'we can assure the reader that any other mathematical structure that derives entropy with minimal assumptions will have at least that many, and usually more'. No paper I know attempts to do anything of the sort.

* Remark 46 *

This concerns the usage of the phrase 'the arrow of time' in general.

I am well aware that lots of people like to use that phrase as a stylistic variant of something like 'the Second Law of thermodynamics', or perhaps of some particular facet of that law. (The first sentence of the abstract suggests that the author uses it that way.) The problem is that sometimes, some authors imply more than that, and this can lead to endless confusion. In fact, anyone tempted to use that phrase should first read this: https://arxiv.org/abs/physics/0402040 .

I hope the author removes all mention of 'the arrow of time'. If not, I would insist that the author insert an explanatory note in the text (not just the abstract), saying something like: 'In this paper, by "arrow of time" I will always mean …', followed by a clear and precise explanation of what the author actually means.

* Remark 47 (the introduction) *

In view of the previous two remarks, I urge the author to rewrite or simply drop the first paragraph and the first sentence of the second paragraph, so that the new introduction immediately tells the reader what the paper is actually about. The paper's target audience is more likely to find the current introduction annoying than engaging, if for no other reason than because it's making them read what is to them is likely to be trivial. I would say that even if this were a journal with a broad interdisciplinary readership like Nature or Science, the current introduction would still not be appropriate---it is too much like popular science. One only needs to go over to their websites and look at the research articles. One will see that they are all very technical from the start. And besides all this, the paper simply doesn't need such a contrived hook. Its actual result (as I summarized it at the very beginning, and as the author himself summarized it in the conclusion) is more than interesting enough to the target audience.

* Remark 48 (the title) *

At present, the title is so vague and uninformative as to be nearly useless in communicating to prospective readers what the paper is about. Here is a possible rewriting: 'A constraint on the eigenstate thermalization hypothesis from the second law of thermodynamics'. Or, 'Second Law of thermodynamics constrains the form of the eigenstate thermalization hypothesis'. After all, the target audience of this paper is assumed to be familiar both with the ETH and the Second Law.

** typos, grammar, and English usage *

p. 1 In This phenomenon has a beautiful statistical reasoning. the word 'reasoning' is inappropriate. It should be something like 'explanation'.

… compare to the number… should be … compared to the number…

… ETH is a criteria… should be … ETH is a criterion…

p. 4 Equation (3) should be preceded by some text such as 'In general, we have that'.

… is the Kelvin’s form… should be … is Kelvin’s form… (or else 'is the Kelvin form')

p. 5 … we will provide numerical evidences… should be … we will provide numerical evidence… ('evidence' is normally an uncountable noun; among the rare cases where it is not, basically all of them are in the context of religion)

p. 6 … We consider system of size $L$… should be … We consider a system of size $L$

In … the leading term starting from different energy eigenstates match the thermal … 'match' should be 'matches', or else 'term' should be 'terms'

pp. 7-8

'psuedo-random' should be 'pseudo-random'

p. 8 In … is within the band ${−W/2, W/2}$… the braces should be replaced by either brackets or parentheses. (The braces denote a set consisting of just two values, $-W/2$ and $+W/2$.)

In …for different values of $E$ which are identified in terms of effective temperature. there should be a comma before 'which'.

… this operator satisfy … should be … this operator satisfies …

p. 9 In the caption of Fig. 7, in … Number of site L = 16 and number of fermions N = 7. … 'site' should be 'sites'.

In … higher order terms will be suppressed by exponentials of the entropy… 'will ' should be 'would'.

p. 10

In … the higher order terms are small compare to the leading term… 'compare' should be 'compared'

In We can still reduced the entropy … 'reduced' should be 'reduce'.

Recommendation

Ask for major revision

  • validity: ok
  • significance: top
  • originality: high
  • clarity: good
  • formatting: excellent
  • grammar: excellent

Author:  Nilakash Sorokhaibam  on 2024-09-10  [id 4756]

(in reply to Report 3 on 2024-07-12)

(Reply to referee report 3, dated 10 September 2024)

Dear referee,

Thank you for the detailed insightful report.
Here I will reply mainly to the critical Remark 1. The full reply will be submitted along with the revised manuscript.

If I may summarised remark 1: It means that the side with denser number of states should contribute more to $\Delta E$.

At the back of my mind, I was always thinking there should be this contribution from the denser side of the spectrum. Your calculation brings it out very clearly. As you have speculated this contribution is suppressed because the peak is not sharp. The peak broadens as we increase the system size. The width of the peak is controlled by $s''(\omega)$ ($s(\omega)$ as defined in the referee report) which is inversely proportional to the heat capacity of the system (as defined in Ref.11 between equation 9 and 11). It arises from the Taylor expansion of $s(\omega)$ about the maxima. So for large system size, at fixed temperature, the contribution just because of the denser region of the spectrum is negligible.

Now we have two numerical evidences supporting this analytic result. First, we have studied $e^{s(\omega)}$ for different system size L. Although only L=14 and L=16 are accessible and presentable (L=12 Hilbert space dimension is too small), plot of $e^{s(\omega)}$ is broader for the larger system. The second numerical evidence is that, according to your calculation in remark 1, the running sum for $\Delta E$ as a function of $\omega$ should be $E_n$ dependent, because the peak region will be $E_n$ dependent. But as we can see in Figure 4 (right side figure), the starting point of the growth of the running sum for $\Delta E$ as a function of $\omega$ is not $E_n$ dependent. The different effective temperatures (T=1,2,3,-1,-2,-3) mean different $E_n$'s. So, this also suggest that contribution from the peak of $s(\omega)$ is quantitatively small.

The artificial numerical exercise as pointed out in Remark 2 is computational very costly because the flipping (or rescaling) has to be done for a large swath of the matrix elements of O. As we have seen, the f-function appears as $f(\bar{E}=E_n+\omega/2,\omega)$, not just for a fixed $\bar{E}$. But I hope the new analytic result and the supporting numerical evidences make the requirement of "ETH-monotonicity" clear.

many thanks again,
NS

---

## Round 10 · Author Response

List of changes
(1) Extensive changes have been made in the manuscript. It has grown from 12 pages to 30 pages. So, let me refrain from listing all the minute changes made.
(2) The results have been fine-tuned and made precise.
(4) Detailed mathematical derivations and arguments of some of the main results have been added.
(3) The peak of the entropic factor (as pointed out by the referee 3) has been studied in detail. The peak does not play any significant role in large systems.
(2) The results have been fine-tuned and made precise.
(4) Detailed mathematical derivations and arguments of some of the main results have been added.
(3) The peak of the entropic factor (as pointed out by the referee 3) has been studied in detail. The peak does not play any significant role in large systems.

---

## Round 10 · List of Changes

(1) Extensive changes have been made in the manuscript. It has grown from 12 pages to 30 pages. So, let me refrain from listing all the minute changes made.
(2) The results have been fine-tuned and made precise.
(4) Detailed mathematical derivations and arguments of some of the main results have been added.
(3) The peak of the entropic factor (as pointed out by the referee 3) has been studied in detail. The peak does not play any significant role in large systems.
(2) The results have been fine-tuned and made precise.
(4) Detailed mathematical derivations and arguments of some of the main results have been added.
(3) The peak of the entropic factor (as pointed out by the referee 3) has been studied in detail. The peak does not play any significant role in large systems.

---

## Editorial Decision

in_refereeing